# Task-Oriented Systematic Design of a Heavy-Duty Electrically Actuated Quadruped Robot with High Performance

**DOI:** 10.3390/s23156696

**Published:** 2023-07-26

**Authors:** Junjun Liu, Zeyu Wang, Letian Qian, Rong Luo, Xin Luo

**Affiliations:** 1State Key Laboratory of Digital Manufacturing Equipment and Technology, Huazhong University of Science and Technology, Wuhan 430074, China; liujunjundhu@126.com (J.L.); D201880208@hust.edu.cn (L.Q.); rong_luo@126.com (R.L.); 2School of Mechatronic Engineering and Automation, Shanghai University, Shanghai 200444, China; htj0585@163.com

**Keywords:** electric quadruped robot, systematic design, mechatronics design, joint design

## Abstract

Recent technological progress is opening up practical applications for quadruped robots. In this context, comprehensive performance demands, including speed, payload, robustness, terrain adaptability, endurance, and techno-economics, are increasing. However, design conflicts inevitably exist among these performance indicators, highlighting design challenges, especially for a heavy-duty, electrically actuated quadruped robots, which are strongly constrained by motor torque density and battery energy density. Starting from task-specific holistic system thinking, in this paper, we present a novel task-oriented approach to the design of such kind of robots, incorporating hierarchical optimization and a control-in-the-loop design, while following a structured design path that effectively exploits the strengths of both heuristic and computational designs. Guided by these philosophies, we utilize heuristic design to obtain the approximate initial form of the prototype and propose a key task-oriented actuator joint configuration, utilizing commercially available components. Subsequently, we build a step-wise analytical models considering trajectory optimization and motor heat constraints for optimization of leg length and joint match parameters to achieve a compact performance requirement envelope and minimize redundancy in the construction of task-specific components. Furthermore, we construct a holistic simulation platform with a module control algorithm for typical scenarios to evaluate subsystem results and adjust design parameters iteratively, balancing conflicts and eventually achieving a reliable design specification for detailed subsystem design. Based on these strategies, we develop a heavy-duty electric prototype achieving a maximum speed of 2 m/s in trotting gait with a load weighting over 160 kg and enduring a period of 2 h. The experiment upon the prototype verifies the efficiency of the proposed approach.

## 1. Introduction

Legged robots have inherent advantages for maneuvering across unstructured terrain due to their isolated footholds that optimize support and traction, as well as the active large-travel suspension afforded by their articulated legs [1]. Over several decades, researchers have devoted significant efforts to improving their performance with an anticipation of wide-ranging applications in industry and daily life. In the early stages, the design and research efforts of quadruped robot teams primarily focused on prototyping platforms. The main objective was to advance the robotic hardware and software technologies, while creating a demonstrative platform to showcase their potential in traversing rugged terrains and exhibiting high-speed stability. However, as the software and hardware technologies continue to evolve and mature, the practical applications of quadruped robots have expanded significantly. Currently, there are commercially available prototypes capable of carrying functional payloads and performing inspection tasks in large-scale industrial environments. Correspondingly, there is a burgeoning demand for quadruped robots that have enhanced comprehensive capabilities, such as higher speed, payload capacity, robustness, terrain adaptability, endurance, and techno-economic efficiencies. These metrics are often constrained by each other due to performance limitations in hardware and software, making it difficult to develop legged robots with comprehensive performance. Hydraulically driven robots such as the BigDog [2] and LS3 exhibit high load and dynamic motion capacity, but their practical applications are limited due to their excessive energy consumption and high noise levels.

Electrically driven robots represent a more environmentally friendly and quiet option, demonstrating high efficiency and simple motor control. During their development, heavy-duty electric quadruped robots that are capable of handling payloads of over 150 kg while maintaining other high-performance features must contend with the dual constraints of motor torque density and battery energy density, which intensify conflicts among the aforementioned practicality indicators, highlighting design challenges.

Currently, there is a lack of task-oriented system design methods for quadruped robots. On one hand, this is due to the fact that the body design, drivetrain, control, and intelligent decision-making technologies of quadruped robots are still not sufficiently advanced. It is necessary to continuously develop and improve these advanced technologies through the construction of experimental platforms. On the other hand, the application of quadruped robots in practical scenarios is still limited or ill-defined. The development of novel application-specific quadruped robots faces challenges of long development cycles and high costs, often requiring the search for suitable application scenarios based on existing platforms. As a result, there is a lack of broad application-driven and economically supported research and development of task-oriented quadruped robots. However, as electrically driven quadruped robots are on the verge of widespread commercial applications, we believe that there is an urgent need for research on task-oriented system design methods for quadruped robots. Specifically, research on electrically driven quadruped robots with high payload and dynamic performance tasks would greatly expand the application scenarios for quadruped robots.

To smoothly develop an electric quadruped robot with a high load and dynamic performance equivalent to the LS3, whilst maintaining high energy efficiency and quiet operation, as well as duly considering techno-economic aspects for commercial applications, we must rely on both heuristic design inspired by human intelligence and computational design. Inspired by the designs of traditional mechatronic complex systems [3,4], we distill the concept of systematic design, which primarily encompassing systems thinking and structured system design methods, and emphasize the inspiration of cascade optimization thinking and control-in-the-loop thinking. Starting from the proposed task, we categorized the entire system into multiple subsystems. The principle of division is to fully leverage the strengths of human creativity and computer numerical computation and reduce the coupling between each subsystem. Based on the characteristics of each subsystem, we decide whether to use computer optimization or human decision making. In the core joint design phase, we propose a task-oriented joint configuration and implement the required functions with commercial components. Subsequently, these subsystems are combined for simulation iteration to further coordinate conflicts in the design variables of each subsystem. When building the overall simulation, considering the complementary relationship between control algorithms and hardware entities, we modularly integrate the advanced MPC+WBC motion control algorithms into simulation platform. This ultimately forms a precise, feasible and detailed design specification to guide the integration of physical prototypes. Based on these strategies, the first generation of electric quadruped robots successfully achieved a load of 179 kg and could reach a speed of 2 m/s in trot gait on grass, and maintain over 2 h endurance at its regular speed. The preliminary estimated COT was 0.52, verifying the effectiveness and efficiency of the proposed method.

The key contributions of our study are as follows: (1) We introduce a task-oriented systemic design methodology for a heavy-duty electrically actuated quadruped robot with high performance, integrating cascade optimization and control-in-the-loop design concepts. This iterative optimization process encompasses task definition, requirement analysis, and core component model optimization, culminating in a comprehensive electromechanical simulation to guide detailed design implementation. (2) We propose novel task-oriented high-torque-density joint configurations with slightly higher gear ratios using commercially available components and validate the effectiveness of this configuration through experiments implementing proprioceptive force control [5] as discussed in Section 6. (3) We use commercial components and open-source software in the whole-machine design process, significantly improving the techno-economic viability and reproducibility of our method. (4) We successfully design, construct and conduct testing on the prototype. Experimental results demonstrate that our approach improves load-bearing performance while maintaining high movement speed and power efficiency, which notably enhances the practical application value of the robot.

The structure of this paper is as follows: Section 2 provides a brief overview of related research. Section 3 briefly introduces the design philosophy followed by the design, Section 4 introduces the analysis model of key design parameters, Section 5 outlines the platform design, Section 6 provides the whole machine experimental results, Section 7 discusses the design method further, and Section 8 summarizes the entire paper.

## 2. Related Research

Designing quadruped robots for specific practical tasks often requires considering comprehensive performance factors such as dynamic motion capability, load capacity, endurance, and robustness. Most current quadruped robot researches tend to concentrate on single or small parts of the aforementioned performance, for instance, speed (MIT Cheetah 2 (21.6 km/h) [6], wildcat (32 km/h) [7]), energy efficiency (MIT cheetah 3 COT: 0.45 [8]), terrain adaptability (Anymal [9,10], mini cheetah [11], spot [7], A1 [12]). Although these typical robots exhibit outstanding performance in speed, energy efficiency, or terrain adaptability, their limited load capacity greatly hinders their abilities in practical scenarios.

Currently, there is no comprehensive and systematic design method available for developing electrically-driven quadruped robots with high payload capacity and dynamic motion performance. However, it is worth noting that successful designs of quadruped robots often involve the combination of the following methods: (1) biomimetic inspiration: developing prototypes based on the morphology, joint trajectories, and foot force research of humans or animals, such as cheetahs [13], kangaroos [14], etc. (2) Simulation-driven design based on simplified [15] or complete models [16]. (3) Leveraging early robot prototypes and iterative development experience, such as HyQ2Max [17], cheetah series [6,8,11], and the Boston Dynamics series [7]. Marc Raibert’s design strategy summary of “Build it, break it, fix it” offers insightful perspectives.

We classify the aforementioned three methods as heuristic-driven design approaches. Based on these methods, innovative advanced component/subsystem design approaches/principles have emerged, such as lightweight, low-inertia leg structures; high-torque quasi-direct drive (QDD) joints [5,18], hydraulic actuators integrated with valves and SEA joints [19] for high-torque and foot-ground dynamic interaction needs; highly adaptable and robust control algorithms (Model Predictive Control (MPC)+Whole-Body Control (WBC) [20], and reinforcement deep learning algorithms). These discrete and vague design principles and strategies provide valuable guidance for the legged robot research community, but their reliance on rich engineering experience to iteratively customize specific advanced components is time-consuming and costly.

(4) Computational design: a promising approach that aligns closely with a systematic design mindset. Several studies have already applied computational design to coordinate multiple conflicting design elements in the robot design process for specific task requirements [21,22,23]. This method parametrizes the robot’s structure and dynamic parameters, and under certain constraints, formulates an optimization problem to find the best solution, which can reduce ambiguous decision-making regarding non-intuitive design variables and laborious manual adjustments, thereby minimizing iterations in prototype development for achieving high-performance and/or superior designs. As early as 1996, Alexander [24] proposed the use of optimization techniques to describe animal morphology and movement. General classification—the computational design—is mainly divided into gradient-based [22,25,26] and gradient-free methods [27], proving a useful strategy in obtaining optimal design parameters for targeted components/systems.

Despite the goal of providing an end-to-end design approach in optimization-based computational designs, the complexity arising from high dimensionality and nonlinearity of design variables, as well as our limited understanding of motion mechanisms, results in difficulties in modeling or resorting to simplified models. The successful implementation of existing computational designs often comes at the cost of constraining the design space, for instance, optimizing designs with given motion trajectory inputs or utilizing existing prototypes/actuators. In other words, this strategy still relies on the preliminary designs of humans, thereby shrinking the scope of the optimization problem.

## 3. Design Philosophy for High-Performance Leg Systems

The design process is fundamentally a series of consecutive decision-making activities, with the evaluation criteria possibly varying based on perspective. Our goal is to develop an electric quadruped robot, within constraints of development cycle and budget, whose payload and dynamic performance are comparable to the LS3. Figure 1 displays the walking speed, self-weight, and payload capacity of typical quadruped robots, all of which fall within the size range of 2 m × 1 m. Most robots exhibit self-weight, payload, and speed capabilities of 100 kg, 20 kg, and 10 km/h, respectively. Only the hydraulically driven BigDog and LS3 showcase significant payload capacity.

Specifically, our objective is to design and develop an electric quadruped robot, comparable in size to the LS3, within a one-year time frame and under a constrained budget. This robot should achieve a payload-to-self-weight ratio of over 0.5, a walking speed exceeding 2 m/s when fully loaded with 180 kg, and a running time of over 1.5 h when fully loaded. We aim to explore the limits of electric quadruped robots and enhance their practical value. As far as we know, no existing research literature has developed an electric quadruped robot with the aforementioned performance. The need for extreme performance necessitates the use of extreme design methods. As such, we summarized the following design philosophies to crack this design conundrum.

### 3.1. Task-Oriented Systematic Design

Our proposed task-oriented systematic design is a system thinking on the one hand, and a structured system design method on the other. System thinking emphasizes viewing the system from a holistic perspective, where the system’s behavior and performance are determined/generated by the dynamic interactions among its constituent parts [4,28]. This study involves multidisciplinary knowledge and relies on team collaboration. We not only consider the physical robotic system but also the influence of the organizational system composed of humans. Under constraints such as funding, cycle, team theory, etc., project success requires targeted support from the management system to enhance the efficiency of robot design and the creative solution of complex problems as much as possible.

In terms of system design methods, we emphasize a task-based systemic design, mainly because most previous quadruped robot development processes lacked clear task indicators. Instead, they were based on biomimetic reference and human creative inspiration, with the aim of achieving ultimate robot performances. This open-ended development process means that many vague decisions lead to these designs with, which exacerbates the difficulty of system/component development and veers design outcomes away from their objectives, leading to a time-consuming and laborious iterative optimization path of multiple prototype versions.

Starting from the proposed task indicators, we organized a design team comprising project managers, domain-specific researchers, and engineers. The team members, administrative management department, sponsors, suppliers, manufacturers, and domain-specific experts coordinated on the assigned tasks and corresponding requirements and held frequent and brief design discussions. The principle of these discussions is democratic; therefore, both hard and soft system methods are equally important [4]. The design process greatly utilized GIGA-Mapping [29], 3D/2D models, simulations, and optimization calculation results. In these design discussions, disagreements are inevitable due to differences in stakeholders’ experiences, design concepts, knowledge accumulation, and positions. When disagreements arise, we require everyone to articulate their views as accurately as possible, conduct frequent discussions with relevant field personnel, and carry out computer analysis and scale model modeling verification if possible. At the agreed time node, a decision must be made to push forward. As our understanding of the design objective deepens with the progression of the design, design discussions gradually become optimized in terms of iterations. Through such efficient frequent and short discussion mechanisms, brainstorming methods, and co-creation, a solution pool for the robot system design was formed, as illustrated in Figure 2a.

Regarding the method for robot system design, as illustrated in Figure 2b, we drew from the V-model [30], starting from the proposed task indicators and following the process of requirements analysis, conceptual design, detailed design, single-leg prototype verification, and whole-machine verification. The requirement analysis translates the tasks into functional and performance needs for a system. The complexity of design stems from the fact that we are not completely certain at the beginning of the functions and exact system/component performances that are needed to achieve the task objectives, as well as conflicting functions and extreme performance requirements. We proposed abstract requirements for the top-level model, including system-wide lightweight properties, high output force and high torque density of joints, safety–controllable–sensible ground interaction force, high rigidity and high load-carrying performance of the body, light-weight low inertia and high reliability of the leg–foot structure, large workspace with concise and reliable performance of joint–transmission structure, high integration and low development cycle, etc. During the conceptual design phase, to further scale down the design elements and shorten the development cycle, we adopted a combination of artificial and computational design methods based on identifying system characteristics. During the computational design phase, we adopted a gradual strategy, starting from a simple model and progressively moving towards more complex models for subsystem/component optimization. In the critical joint design, we considered the impact of motion trajectory and a control algorithm on joint performance requirements. Starting with trajectory optimization, we then carried out a leg length optimization to acquire the torque–speed performance requirements of the joint. Subsequently, we proposed an innovative task-oriented joint drive configuration scheme, optimizing the design of joint configuration within a candidate set of commercial motors and reducers. Throughout the design process, there was iterative interaction between artificial design decisions and computational design. This interaction allowed initial values and constraints to be updated, and other design features could be determined. Finally, the control algorithm module of the physical prototype was embedded in the whole-machine simulation to further iterate and update design parameters, verify the rationality of requirements, and seek feasible or optimal solutions to conflicting requirements. This ensured that design results were further optimized on the basis of meeting standards and reliability, as well as output specifications, to guide detailed design.

### 3.2. Target Cascading Optimization Approach

The concept of objective cascade optimization systematically decomposes large-scale systems into multiple subsystems, with the overall objective further divided and passed down to each subsystem. Each subsystem is designed and optimized independently, and then combined to ascertain if the composite system achieves the target objective. Within this framework, the entire system is divided into the following subsystems based on their functions and in adherence to the principle of minimizing inter-system coupling: morphological size system, control system, and energy system. The morphological size system is further divided into joint system, leg-foot system, and body framework, where the design of a brand new, high-performance quadruped robot is an activity that heavily involves innovative intelligence (Figure 3). In the face of complex systems with multiparametric coupling, computational design strategy, which truncates the design space, can only offer guidance on some design parameters. Innovative design heavily leans on human wisdom, especially the superior intellectual processes such as imagery, intuition, and inspiration, whereas the intelligence realized by computers comprises only the logical and computable aspects of human intelligence. Building on the foundation of this systemic hierarchy, our study emphasizes the amalgamation of human cognition and computational intelligence, suggesting that both software/hardware system approaches hold equal importance in design optimization. As Roger Martin pointed out, reliance solely on analytic or intuitive thinking is insufficient to maintain a competitive advantage, as each approach, while providing substantial strength, can also generate systemic weaknesses if applied in isolation.

Based on the characteristics of the subsystems, we construct analytical models and optimize them using computers, or make manual decisions based on the designers’ insights, taking full advantage of the creativity of humans and the numerical computation capabilities of computers. The subsystems designed by manual decisions include control systems, motor drive systems, and empirical designs of non-critical components of the body and the initial configuration/parameters design of various subsystems. The outcomes of manual decisions serve as inputs for computational design, which is then optimized by computers to achieve optimal leg size and driving joint performance requirements, further guiding the optimized selection of driving joints. Beginning with the abstract top-level model, we gradually refine it into a detailed digital prototype model, and then perform control-in-the-loop simulations on the entire machine to validate the design of each subsystem, and iterate optimization. This approach significantly reduces the dimension and complexity of design problems, shortens the research and development cycle, and allows outsourcing the design of specific subsystems/components to suppliers (for example, batteries and energy management systems are provided by specialized manufacturers based on the performance parameters we provide). In addition, due to the modular nature and loose coupling of this hierarchical optimization framework, it is straightforward to extend the design parameters involved in the computational design, such as the length of the linkage in the knee joint transmission chain, the number of leg segments, the stiffness of the parallel springs, and the layout. This allows us to reduce some heuristic decisions and achieve an optimized design from another perspective.

### 3.3. Control-in-the-Loop Design

The improvement in the locomotion performance of legged robots relies on two key technological breakthroughs. Firstly, it relies on the development of more robust and applicable motion control algorithms based on a deeper understanding of foot motion mechanisms. Secondly, it also relies on the integration of a driving–sensing–dynamic–computing prototype platform with enhanced performance. These two technologies complement each other and drive mutual progress [31]. For example, the SLIP-inspired motion control concept guides the direction of design towards lightweight legs, high-bandwidth driving joints, or the addition of passive spring mechanisms to mimic specific spring stiffness characteristics [32]. Similarly, the concept of motion control based on the ZMP stability principle leads to the design of wide feet in bipedal robots to ensure that the center of mass falls within the support polygon formed by the foot placements, while quadruped robots often adopt a static walking gait.

The initial phase of control-in-the-loop design involves determining the control strategy route before the design process commences. This approach allows for the design of corresponding mechatronic structures that align with the characteristics of the control algorithms, reducing discrepancies between mathematical models and physical prototypes. Consequently, the performance of the digital prototype accurately reflects that of a real robot, while also facilitating the consideration of component-level limitations and avoiding redundant designs that exacerbate the difficulty of extreme designs.

We adopted the MPC+WBIC controller. This approach segregates motion control into two simpler controllers. The first controller formulates the whole mechanism as a simple lumped mass model and uses Model Predictive Control (MPC) to establish the optimal distribution of reactive forces along an entire gait motion cycle; the second controller employs a whole-body dynamics model and high-frequency feedback control to achieve high-bandwidth control, generating more accurate torque commands than the lumped mass model. This necessitates that the leg mass proportion of the physical prototype we design is as small as possible (less than 10% empirically) to effectively approximate the mapping to the lumped mass model. Furthermore, for successful execution of such a complex control algorithm, it is mandatory that the motion control system has enough computational capacity, and the leg joint system possesses necessary output accuracy and output force bandwidth.

The second phase of control-in-the-loop design is software-in-the-loop simulation, facilitating high-precision simulation-driven methodology [15,16,33,34] for the design. This phase involves evaluating and fine-tuning the performance of the control algorithms, improving structural designs, and obtaining updated and more reliable joint velocity-load profile curves to guide the iterative optimization of each subsystem. This approach further helps to reduce the gap between digital and real prototypes, promoting the discovery of breakthroughs or feasible balance in situations of intense conflict among multiple design elements in high-performance complex electromechanical system design.

## 4. Analytical Model for Key Design

The locomotion performance of a legged robot heavily relies on the design of its joints and legs, as the motion is dependent on the interaction forces between the leg and the ground driven by the joints. Traditional manual design methods often suffer from uncertainty, complexity, and time consumption. To ensure the design results converge to the desired objectives and avoid excessive performance redundancy, this section introduces how to optimize the design using appropriate analytical models. Key design decisions include morphological parameters such as leg degrees of freedom, number of leg segments, segment lengths and proportions, leg configuration and layout, as well as the design of driving joints. For motor-driven joints, it involves the configuration and optimization design of motors and reducers.

Typical mathematical optimization problems can be formulated as follows:(1)$$\begin{array}{c} \min f(X)\\ \;s.t.\;\left\{\begin{array}{cc} g_{i}\left(X\right)\geq 0 & i=1,2,\ldots ,m\\ h_{j}\left(X\right)=0 & k=1,2,\ldots ,l \end{array}\right. \end{array}$$
where, $X\in R^{n}$ represents a vector composed of decision variables, $f:R^{n}\to R$ is a scalar objective function, $g_{i}:R^{n}\to R^{m}$ denotes the inequality constraint equations, and $h_{j}:R^{n}\to R^{k}$ signifies the equality constraints. If any among *f*, $g_{i}$, or $h_{j}$ is a nonlinear function, the formulation as per (Equation 1) constitutes a nonlinear programming problem. Numerous pre-existing algorithms can directly solve such problems, for instance, the MATLAB’s *fmincon* function.

Guided by the design principles discussed in Section 3, we started with task objectives and used a simplified centroid dynamics model to optimize the centroid trajectory and foot placement for the given task. Then, we employed an eight-DoF whole-body dynamics model for optimization to obtain optimal mechanical performance of joints and leg segment lengths. Based on these results, suitable motors and reducers are selected via joint-level optimization. Finally, a detailed control-digital prototype simulation model is built to validate and refine key design variables, as shown in Figure 4.

### 4.1. Morphological Optimization Design

For a specific task, the different morphological features of a robot have varying demands in driving performance. To address the proposed performance objectives, a mathematical model is developed for optimizing morphological features. Initially, a simplified 2D single rigid-body model was employed with four variable nominal leg lengths to simulate foot–ground interaction points and optimize the motion trajectory for a given user-defined task, as shown in Figure 4A. The optimization problem can be expressed as follows:
(2)$$\begin{array}{cc} \mathrm{Find}\; & r\left(t\right)\in R^{2},\;\theta \left(t\right)\in R,\;T\in R\\ \; & P_{i}\left(t\right)\in R^{2},\;i\in N\\ \mathrm{Target}:\; & \min C_{1}=w_{1}|\theta \left(T\right)-\theta \left(0\right)|+w_{2}\left|\dot{\theta }\left(T\right)-\dot{\theta }\left(0\right)\right|\\ \; & \;\;\;+w_{3}\sum_{i=1}^{N}\int_{td,i}^{to,i}\left|X_{h,i}\left(t\right)-P_{i,x}\left(t\right)\right|dt\\ s.t.\; & r_{\mathrm{init}},\theta_{init}=\left[r_{0},\theta_{0}\right]\\ \; & \dot{X}c=V_{task}\;(\mathrm{task}\;\mathrm{velocity})\\ \; & \mathrm{for}\;\forall t\leq T,i\in N:\\ \; & G_{g}=s_{i}\left(t\right)\;(\mathrm{gait}\;\mathrm{graph})\\ \; & {\left[\ddot{r(t)},\dot{\theta (t)}\right]}^{T}=f_{d}\left(r\left(t\right),P_{i}\left(t\right),f_{i}\left(t\right)\right)\;(\mathrm{dynamic}\;\mathrm{model})\\ \; & P_{i}\left(t\right)\in R_{i}\left(r,\theta \right)\;(\mathrm{kinematic}\;\mathrm{model})\\ \; & s_{i}\left(t\right)P_{i,y}\left(t\right)=0\;(\mathrm{flat}\;\mathrm{terrain})\\ \; & \left|(1-s_{i}\left(t\right)\right)\left|f_{i}\left(t\right)\right|=0\;(\mathrm{no}\;\mathrm{force}\;\mathrm{in}\;\mathrm{swing}\;\mathrm{phase})\\ \; & s_{i}\left(t\right){\dot{P}}_{i}\left(t\right)=0\;(\mathrm{no}\;\mathrm{slip})\\ \; & \left|f_{i}\left(t\right)t\left(P_{i}\right)\right|<\mu f_{i}\left(t\right)n\left(P_{i}\right),\;\mathrm{if}\;s_{i}=1\;(\mathrm{friction}\;\mathrm{cone})\\ \; & \left|r\left(t\right)-P_{i}\left(t\right)\right|\leq l_{\max}\;\\ \; & f_{i}\left(t\right)n\left(P_{i}\right)\geq 0,\;\mathrm{if}\;s_{i}=1\;(\mathrm{pushing}\;\mathrm{force}) \end{array}$$$r\left(t\right)\in R^{2}$: 2D vector, representing the position of the robot’s center of mass (CoM) in the horizontal plane. θ(t)∈R: Scalar, representing the orientation angle of the robot in the plane. T∈R: Scalar, representing the total time of trajectory optimization. $P_{i}\left(t\right)\in R^{2}$: 2D vector, representing the position of the robot’s *i*-th foot in the horizontal plane. i∈N: Integer, representing the index of the robot’s feet (1 to *N*, where *N* is the total number of feet). $C_{1}$: Objective function, aimed at minimizing the robot’s orientation change, angular velocity change, and the distance between the foot tip and the desired trajectory. $w_{1},w_{2},w_{3}$: Weight coefficients, used to adjust the priority of different terms in the objective function. $r_{\mathrm{init}},\theta_{init}$: Initial CoM position and orientation angle of the robot. $V_{task}$: Task velocity, representing the desired velocity of the robot during the trajectory execution. $G_{g}$: Indicator function for the i-th foot at time *t*, representing the stance phase or swing phase (1 for stance phase, 0 for swing phase). $f_{d}\left(r\left(t\right),P_{i}\left(t\right),f_{i}\left(t\right)\right)$: Dynamic model, representing the relationship between the robot’s CoM position, foot tip position, and foot tip force. $R_{i}\left(r,\theta \right)$: The reachable space of the *i*-th foot, given the robot’s CoM position and orientation angle. $s_{i}\left(t\right)$: At time *t*, indicator function of the *i*-th foot’s stance/swing phase (1 for stance phase, 0 for swing phase). $f_{i}\left(t\right)$: Vector, representing the force acting on the *i*-th foot at time *t*. μ: Friction coefficient, representing the frictional force between the foot tip and the ground. $l_{\max}$: Scalar, representing the maximum extension length of the robot’s leg.

The inputs to the system include prior human knowledge, such as the body mass, inertia, standing height, as well as gait parameters and force profiles of foot-ground contact for specific tasks. The continuous-time trajectory optimization problem is discretized into *N* points with equal time intervals δT, and the walking period is denoted as *T*. The centroid, orientation, foot positions, and contact forces are represented in Cartesian coordinates. The optimization objective is to achieve periodic stable motion under given constraints, as indicated by the first two terms of the objective function. Additionally, we aim to minimize the moment arm of foot placement *i* relative to the hip joint position xp during the entire support phase (from touchdown time td to take-off), effectively minimizing the joint torque. The constraints include the given initial state and target velocity *v*, as well as the gait phase diagram (with a 50% duty cycle for trot gait). To ensure physical feasibility, a simplified 2D centroid dynamics model is employed, represented by the following equations:
(3)$$\begin{array}{cc} M{\ddot{X}}_{C}\left(t\right) & =\sum_{i=1}^{N}f_{x,i}\left(t\right)\\ M{\ddot{Y}}_{C}\left(t\right) & =\sum_{i=1}^{N}f_{y,i}\left(t\right)-Mg\\ I\ddot{\theta }\left(t\right) & =\sum_{i=1}^{N}s_{i}f_{i}\left(t\right)\times \left(r\left(t\right)-p_{i}\left(t\right)\right)=\tau_{\theta }\\ & =\sum_{i=1}^{N}s_{i}\left[\left(P_{i,x}-X_{C}\right)f_{y,i}-\left(P_{i,y}-Y_{C}\right)f_{x,i}\right] \end{array}$$

Here, *M* is the robot mass, $X_{C},Y_{C}$ represent the centroid position in the *x* and *y* directions, r(t) is the centroid position vector, *N* is the number of legs, g denotes the gravitational acceleration, $f_{i}$ represents the total force at the *i*-th leg’s foot, $f_{x},f_{y}$ are the foot forces in the *x* and *y* directions, $p_{i}$ represents the position of the *i*-th leg’s foot, and *I* denotes the calculated moment of inertia. During the support phase, the input foot force approximates a semi-sinusoidal curve, which must satisfy the constraints of momentum conservation and friction cone. The foot force profile can be parametrized using a fifth-degree Bezier polynomial, and the polynomial coefficients are incorporated into the optimization decision variables. Taking a cue from the semi-sinusoidal foot contact force profile characteristic of a dog’s walking pattern, we adopted a simplified deterministic semi-sinusoidal force input scheme.

Subsequently, we input the optimized trajectories and prior knowledge from humans (for instance, the use of a two-segment leg to ensure a simple and reliable structure; a mammalian configuration to reduce the overall width of the body, efficient utilization of leg length to increase body height and enhance static off-road capability; an O-shaped layout primarily to free up cargo area in the middle of the body and structurally reduce the interference range between the front and increase rear legs; leg segments of equal length for maximizing the workspace to improve static off-road capability; and coaxial design of hip and knee drive joints to decrease the mass and inertia of the linkage leg) into a second eight-degree-of-freedom whole machine dynamics model, as shown in Figure 4B. We made a thigh-calf model with specific length and mass inertia parameters reproduce the optimized trajectory of the previous model. The length of the linkage is the decision variable, which is used to achieve optimal comprehensive performance. The optimization formula is as follows:(4)$$\begin{array}{cc} \mathrm{Find}\; & L_{1},L_{2}\in R\\ \mathrm{Target}:\; & \min C_{2}=\sum_{j=1}^{8}(w_{4}Norm\left(|rms\left(\tau_{j}\right)\right|)+w_{5}Norm(|rms\left(power_{j}\right)\left|\right)\\ \; & +w_{6}Norm\left(|\left(max\left(\omega_{j}\right)|\right)+w_{7}Norm\left(L_{i}\right)\right)\\ s.t. & M\left(\chi \right)\ddot{\chi }+C\left(\chi ,\dot{\chi }\right)+G\left(\chi \right)=S^{T}\tau +J{\left(\chi \right)}^{T}f\;(\mathrm{dynamic}\;\mathrm{model})\\ \; & \chi ={\left[X_{C},Y_{C},\theta ,q_{j}\right]}^{T},\tau ={\left[\tau_{1}\cdots \tau_{8}\right]}^{T},j\in 1\cdots 8\\ \; & P_{i}\left(t\right)=R_{i}\left(X_{C},Y_{C},\theta ,q_{j}\right)\;(\mathrm{kinematic}\;\mathrm{model})\\ \; & L_{1}+L_{2}\geq l_{hp}\\ \; & L_{1}=L_{2}\\ \; & \left\{\begin{array}{cc} \left|rms\left(\tau \right)\right|\leq \tau_{c} & \\ \left|max\left(\tau \right)\right|\leq \tau_{max}\;(\mathrm{motor}\;\mathrm{constraints}) & \\ |max\left(\dot{q}\right)|\leq \omega_{max} \end{array}\right. \end{array}$$
where $L_{1},L_{2}\in R$: Scalars, representing the lengths of the two segments of the robot’s leg. $C_{2}$: Objective function, aimed at minimizing the comprehensive performance of torque, power, maximum angular velocity, and leg length for all eight joints. $w_{4},w_{5},w_{6},w_{7}$: Weight coefficients, used to adjust the priority of different terms in the objective function. $rms(\tau_{j})$: Root mean square value of the torque for the *j*-th joint. $power_{j}$: Power of the *j*-th joint. $max(\omega_{j})$: Maximum angular velocity of the *j*-th joint. $L_{i}$: Length of the *i*-th leg segment. $\chi ={\left[X_{C},Y_{C},\theta ,q_{j}\right]}^{T}$: Vector, representing the robot’s position, orientation, and joint angles. $\tau ={\left[\tau_{1}\cdots \tau_{8}\right]}^{T}$: Vector, representing the torques of the eight joints. M(χ): Mass matrix of the robot. $C(\chi ,\dot{\chi })$: Coriolis force matrix. G(χ): Gravity matrix. $P_{i}\left(t\right)$: Position of the *i*-th foot in the horizontal plane. $R_{i}\left(X_{C},Y_{C},\theta ,q_{j}\right)$: Reachable space of the *i*-th foot, given the robot’s position and orientation. $l_{hp}$: Minimum length that the robot’s leg needs to satisfy. $\tau_{c}$: Allowed torque capacity for the joints. $\tau_{max}$: Allowed maximum torque for the joints. $\omega_{max}$: Allowed maximum angular velocity for the joints.

In this configuration, the eight leg segments across four legs are designed to have equal proportions. The estimated distance from the hip joint to the CoG is 0.65 m, and the estimated masses of the hip and knee joints are 15 kg, respectively. These parameters were initially estimated roughly from experience. After trajectory optimization, the rotation angles and driving torques of the hip and knee joints can be derived through inverse kinematic and dynamic solutions. Subject to constraints such as peak motor torque, rated torque, and peak speed performance, we aim to find the optimal leg length to minimize the cost function, considering four comprehensive normalized performance demands including root mean square torque, root mean square power, maximum speed, and leg length. The final result, L1 = L2 = 0.5, is achieved after the optimization iterations depicted in Figure 4C. Figure 5 illustrates the torque-velocity performance requirements for the hip and knee joints of the hind leg, obtained by optimizing the leg length in the eight-DoF (considering only the leg DoF) floating-base dynamic model (Figure 4B) using Equation (Equation 4). The table in the Figure 5 presents the root mean square torque of the hip and knee joints on the hind side, which closely matches the root mean square torque obtained from the overall system simulation (as show in Table 1). These requirements further serve as preliminary performance constraints for the optimization and matching of joint motors and reducers (Equation (Equation 5)), aiding in the identification of an initial set of joint drive combinations to guide detailed structural design and subsequent overall system optimization.

### 4.2. Modular Actuators Desgin

Joint design is of vital importance to the development of legged robots. When choosing the drive configuration of the motor and reducer, the selection of the appropriate reduction ratio is key. Conventional motors operate at high speeds, but their output torque is relatively low. Ignoring losses, a gearbox can amplify a motor’s output torque by a factor of *N*, while simultaneously decreasing its output speed by the same factor. The load rotational inertia perceived by the motor is in fact the load rotational inertia divided by the square of the reduction ratio, i.e., $J_{L}/N^{2}$. In specific conditions, from the perspective of the motor as an energy input, the reduction ratio directly affects the required motor output torque. A high reduction ratio means a motor with a lower output torque is required. Accordingly, a lower output current is required for the motor driver, reducing associated hardware costs, and a less precise encoder can be used at the motor end. The optimal reduction ratio for the minimum motor output torque requirement is $N=\sqrt{J_{L}/J_{m}}$.

However, from the perspective of the energy input caused by foot–ground inter-actions, as the reduction ratio increases, friction damping in the transmission structure tends to increase, and the motor inertia translated to the joint output end is $J_{m}*N^{2}$. When energy is conveyed from the foot to the joint output end, the mechanical impedance increased (high reduction ratios reduce back-drivability), causing an increase in foot–ground contact collision force at equivalent landing speeds. Moreover, achieving high-precision foot force control necessitates complicated control algorithms or force sensor support. Therefore, from the standpoint of structural and control algorithm simplicity and safe, controllable foot–ground interactions, the best match is a design that employs direct drive without a reducer. Direct drive has the benefits of high transmission rigidity, high dynamic performance, and low transmission loss. High-precision foot force control can be accomplished via a simple motor closed-loop current torque control. However, the torque density of the joint significantly decreased.

For the design of quadruped robot joints that require a high torque density, a widely used compromise matching method is to use quasi-direct drive joints, i.e., a first-stage planetary reducer with a reduction ratio of less than 10 and a high-torque motor, or to use series elastic joints (SEA), implementing dynamic interaction and force control via a series elastic structure under a high reduction ratio.

Existing robot drive design methods each have their advantages, and the resultant empirical design principles offer valuable insights. However, they may also lead future researchers into conventional traps, obstructing further exploration and trials with new approaches. Given our design objectives, to reduce technical complexity and shorten the development cycle, one of our design principles is to utilize commercial components wherever possible. Traditional quasi-direct drive joints require bespoke designs of ultra-high-torque motors and matching reducers, and SEAs also require the custom design of elastic structures, which add to the complexity of the structure and control methods.

Considering these factors, we revisited the methodology for selecting reduction ratios. More specifically, a key question for designers is: can we accomplish the three stated objectives by utilizing commercially available reducers that feature mature manufacturing processes (such as two-stage planetary, harmonic, and RV reducers, with speed ratios typically exceeding 10)? This could lessen the stringent performance requirements of motors, thus enabling the use of commercial motor drivers and their related components, thereby accelerating the R&D process and reducing costs.

Based on the positive hypothesis of the aforementioned questions (a detailed analysis is beyond the scope of this paper), we preliminarily selected a suitable set of commercial motors and reducers using the static information in the manufacturer’s directory and empirical rules, according to the joint performance requirements obtained from Section 4.1. Considering the high dynamic activity of heavily loaded quadruped robots, the joint torque experiences substantial fluctuations and peak torques. The aforementioned selection methodology may result in performance redundancy or overheating due to insufficient drive capacity. To pinpoint a combination of motor and reducer that can most closely envelope the performance requirements, we introduced thermal constraints for the motors, outlining the specific optimization methods as follows.
(5)$$\begin{array}{cc} \mathrm{Find}\; & \;\mathrm{Motor}+\mathrm{gear}\;\mathrm{combination}\;\mathrm{in}\;\mathrm{candidate}\;\mathrm{set}\\ \mathrm{Target}:\; & \min C_{3}=Weight\left(motor\right)+Weight\left(gear\right)\\ \; & s.t.\;\mathrm{Motor}\;\mathrm{constraints}\\ \; & \left\{\begin{array}{cc} \left|max\left(\tau \right)\right|\leq \tau_{max}\; & \\ |max\left(\dot{q}\right)|\leq \omega_{max} \end{array}\right.\\ \; & \tau_{max}=\left(\tau_{m,p}-\left(J_{m}+J_{g}\right){\ddot{q}}_{l}\;\cdot \;N-T_{\mathrm{fric}}\right)\times N\\ \; & \Delta T_{st}=\left(T_{st}-T_{a}\right)=R_{th}\left(I_{m}^{2}R_{T_{w}}\right)<min\left\{\Delta T_{class},T_{gear}\right\}\\ \; & \mathrm{Gear}\;\mathrm{constraints}\\ \; & \left|rms\left(\tau \right)\right|\leq \tau_{s1}\\ \; & \left|max\left(\tau \right)\right|\leq \tau_{s2}\\ \; & M_{max}\leq M_{c} \end{array}$$
where $\tau_{m,c}$$\tau_{m,p}$ is the actual rated torque and peak torque of the motor, $J_{m}$ is inertia of the motor rotor, $J_{g}$ is the inertia of the reducer, ${\ddot{q}}_{l}$ is the angular acceleration of the joint output, *N* is the deceleration ratio, $T_{\mathrm{fric}\;}$ is the friction torque, and other thermal-related parameters such as ambient temperature $T_{a}$, motor thermal resistance $R_{th}$, motor current $I_{m}$, and motor winding resistance $R_{T_{w}}$. In this optimization problem, we aim to find the optimal motor and gear combination from a set of candidates, subject to motor performance and first-order thermal model constraints, while minimizing the objective function $C_{3}$, which represents the total weight of the motor and gear. The constraints include maximum torque constraint $\tau_{max}$ of the motor, the maximum angular speed constraint $\omega_{max}$ of the joint, and the first-order thermal model constraint instead of empirical root mean square moment constraints, which ensures that the estimated stator temperature rise $\Delta T_{st}$ of the motor does not exceed the smaller value between the motor winding temperature rise limit $\Delta T_{class}$ and the gear temperature rise limit $T_{gear}$. Due to the mechanical properties of the reducer, the starting and stopping torque allowed by the selected reducer should be greater than the root mean square torque of the joint output end. Its maximum instantaneous permissible torque should be higher than the maximum torque of the joint output end. As the reducer’s end directly connects to the load, under the constraint of the reducer’s bearings, the maximum torque $M_{max}$ generated by the load at the reducer’s end face should be less than the reducer’s maximum permissible torque $M_{c}$.

We incorporate the most comprehensive set of commercial motors and reducers possible into our optimization problem’s candidate set, seeking an optimal solution via integer optimization. To narrow down the candidate set and reduce manual collation work for complex product catalogs, we analyzed the mechanical demand characteristics of the joints. We found that when the peak torque of the output end exceeds 1300 N·m, the space for selecting lightweight commercial components to achieve as high a torque density as possible and meet demand characteristics is relatively small. Especially in the choice of reducer, only RV reducers perform effectively in the high-output-torque range exceeding 1300 N·m among RV, harmonic, and planetary reducers. For instance, the harmonic reducer SHF-58-50-2UH, with a performance comparable to that of RV42N-41-B (weighing 6.4 kg), weighs up to 20 kg, and the ADS140 planetary reducer from APEX DYNAMICS weigh 13.5 kg (reduction ratio 4–10) and 16.6 kg (reduction ratio 16–91), respectively.

While satisfying the above constraints, choosing as low a reduction ratio as possible can enhance the transparency of the joint drive, achieving a widely used quasi-direct drive joint configuration (mainly adopting a reduction ratio less than 10). However, a low reduction ratio increases the torque output demand of the motor, especially for the heavy-duty robots we are designing. When choosing the ADS140 reducer with a reduction ratio of 10, the hip-knee joint motor model that basically satisfies the simulation performance requirements for the high-torque-density KBM series motors is KBM-45X02, which weighs 17.5 kg, making the total weight of the motor and reducer for the hip-knee joint alone reach 31 kg. According to the weight distribution of existing electrically driven prototypes, the weight of the joints accounts for over 50% of the total weight, so it is vital to integrate lightweight joints. This necessitates reevaluating the high joint torque density brought about by a higher reduction ratio, reducing the demand for high-torque-density motors, while achieving high transparency, high-precision torque output, low output inertia, and good foot-ground interaction characteristics similar to direct-drive joints.

Based on these needs, we finally choose the RV42N reducer with a reduction ratio of 41 to achieve good foot-ground interaction and force output precision (the detailed analysis is beyond the main content of this paper). This greatly simplifies the task of collating the candidate component set. Once the reduction ratio is determined, we can screen out several groups of candidate motors, and then find the optimal joint match in the given candidate component set through the optimization design method mentioned above, ensuring its performance meets the joint performance requirements proposed in the previous section. At this stage, we can obtain more specific information about the drive performance, dimensions, and weight of the hip-knee joint, achieving a more detailed digital prototype design, as shown in Figure 4C. As an input for the control-whole machine simulation, the final joint optimization design needs to be calibrated after the whole-machine electromechanical simulation, and the joint demand characteristics results of the whole-machine simulation are shown in Table 1. According to the simulation results, we further improve the design of the yaw joint. As the yaw joint is mainly used for turning and lateral speed regulation, it only needs to realize the functions of turning and lateral walking regulation, so the joint characteristics do not need to consider foot-ground interaction, but only high-precision position control. Based on this consideration, we further relax the reduction ratio limit for the yaw joint reducer and choose the RV25-N with a reduction ratio of 81. Then, we return to the optimization process in Formula (Equation 5) and complete the optimized matching of all 12 joints of the entire machine. The design results after iteration are shown in Table 2.

### 4.3. Holistic Simulation Design

Prior to implementing a detailed physical model, a simulation verification of the whole robot model is necessary, combined with the control algorithm for optimization and the correction of design parameters, thereby minimizing the deviation between the digital prototype and the physical one.

#### 4.3.1. Simulation Framework

As illustrated in Figure 6, we adopt the open-source legged robot control algorithm, MPC+WBIC, where gait, step frequency, and duty cycle parameters are referenced from the results of trajectory optimization. The weight parameters in MPC+WBIC and underlying PD parameters are adjusted manually. We construct an integrated simulation platform using the Webots simulation platform. The detailed physical model of the robot is developed in Solidworks and converted to a URDF file to be loaded into Webots. The control algorithm is packaged into a module that the Webots controller module loads into memory. It interacts with the Webots’ ODE engine via explicit linking of the dynamic library. This decoupling design not only facilitates the independent debugging of the control program but also simplifies the direct deployment of the simulated control algorithm to the physical prototype’s controller.

#### 4.3.2. Simulation Result

Following multiple iterations of system simulation, we iteratively refine our design based on the joint drive performance requirements under two operating conditions (2 m/s trot on flat ground with a maximum load of 180 kg and on-the-spot turn at about 1 rad/s). Figure 7 represents the torque-velocity requirements curve obtained through final iterative improve design process (under the final joint configuration, mechanical leg–foot structure, and control parameters). The corresponding root mean square torques and other characteristics are presented in Table 1.

## 5. General Description

### 5.1. Mechanical Design

We developed a lightweight, load-bearing body frame design. The body frame is formed of a main bearing frame and stiffener. The main bearing frame was constructed by bending and welding 316 seamless steel tubes, adopting the topological optimization design method. The total frame weighs 26 kg, and the first-order modal frequency is 15 Hz. As shown in Figure 8B, from front to back, six X-shaped 7075 aluminum stiffeners are attached to the coronal frame plane, and 2 X-shaped 7075 aluminum stiffeners are fastened on either side of the body to strengthen the stiffness of the body, which also serves as part of the installation surface for the equipment. The belly area of the frame is divided into four layers, which (from top to bottom) are the load mounting plane, drivers and controller mounting plane, high-voltage battery pack (output 380 V) mounting plane, and low-voltage battery pack (output 5 V 24 V) mounting plane.

The drivers are symmetrically positioned and close to the actuated motors, which minimizes the length of the high current power line to reduce EMI impact. The single leg connecting the flange and frame is fastened together with bolts, making the installation and maintenance of a single leg fast and straightforward.

The lightweight, low-inertia leg design increases open-loop force control bandwidth while decreasing impact. The simple transmission chain and load-carrying structure present strong mechanical robustness. Following the above principles, the leg features a two-segment (0.5 m long) and three-degree-of-freedom (similar to mammal leg structure, hip abduction/adduction (HAA), hip flexion/extension (HFE), and knee flexion/extension (KFE)). Table 3 shows detailed parameters. The knee joint drive module lifts and actuates the knee joint using a four-bar linkage, and the three joints’ rotation axes cross. Alloy steel is utilized for axle components, 7075 aluminum alloy is used for all other leg structural elements, and the total weight of the leg structure is 8 kg except for the actuation joint, which can withstand a vertical force of more than 4000 N (natural standing posture, nominal leg length is set to 0.8 m). The approximately elliptical foot structure is covered in a 15 cm thick hollow rubber pad to enhance impact resistance. The four-link drive achieves KFE 10–150 degrees rotation range, HAA rotation range is 45–120 degrees, and HFE rotation range is 20–170 degrees (as shown in Figure 8B), realizing a large foot-end working space, which is capable of increasing the robot’s adaptability to different terrain and obstacle-crossing ability. Table 2 displays the configuration parameters of each joint.

### 5.2. Hardware Architecture and Communication

Figure 8C shows the hardware and communication architecture. The NUC locomotion controller (i7 @ 4.7 Hz, 16 G RAM), which runs the Linux operating system, realizes sudo-realtime operation with the preempt RT patch, is responsible for motion planning, and sends the expected joint angle and joint torque to the low-level controller via UDP connection at a frequency of 500 Hz. The latter runs PD + feed-forward torque control using TwinCAT software installed on a Beckhoff c6030 and monitors the operational status of the Elmo drives at a frequency of 1 kHz through EtherCAT. The Elmo driver is configured in synchronous torque mode, and the current closed-loop bandwidth exceeds 4 kHz. NVIDIA${}^{®}$ Jetson AGX Xavier${}^{\mathrm{TM}}$, the perception computer, is in charge of lidar environment perception and path planning. It communicates with the NUC via UDP. To establish human-computer interaction, commands and robot status data are exchanged via WiFi between the robot and the portable tablet. The tablet and the operating joystick communicate wirelessly at 2.4 GHz. By sharing memory, the motion control computer and the simulation PC realize the software in the loop test, improving debugging efficiency and security.

## 6. Experiments

According to the design specifications, we first built a single-leg test platform to test joint performance and motion control architecture, and then carried out integrated design of the entire system as shown in Figure 9. The open-source algorithms [20] were deployed in the controllers; we only changed the robot’s physical characteristics, the joint PD controller, gait parameters, and the communication mode, as indicated in Figure 8. Below are brief introductions to several experiments on task indicators.

### 6.1. Speed Test under High Load

Without employing an accurate joint friction dynamics model, the robot can walk on grass at a speed of 2 m/s with a load of 179 kg. This result shows that the joint design paradigm and whole-machine integration scheme are feasible. As depicted in Figure 10a, the robot maintains an average velocity exceeding 2 m/s for a duration of more than 2 s (we consider the maximum average velocity within a 2 s sliding window as the peak velocity achievable by the robot. The image illustrating the analysis of average velocity is provided as a supplementary file, accessible through the Data Availability Statement link). Figure 10b–f illustrate the foot position tracking curve in the world coordinate system, the body posture, and the torque-speed feedback curves of three joints on one leg at the stage of highest speed (highlighted by the dashed box in Figure 10a). Under full load, it can be observed that the robot maintains stable high-speed movements and demonstrates a compact joint performance envelope.

### 6.2. Endurance and Energy Efficiency Test

To evaluate energy efficiency under load conditions, the robot walked for 14 min and stepped for 1 min while carrying a 171 kg payload (total weight: 510 kg). The test was 15 min, the distance traveled was 610 m, and the overall electric consumption was 8.8%. The battery capacity was 6.22 kwh, and the discharge range was calculated as 100–10%.

### 6.3. Other Test

Load test: the robot’s payload was increased to 210 kg, and the robot could walk at 0 velocity. The joint torque is as indicated in Figure 11. Force control accuracy test: we used the motor current loop to achieve open-loop force control. The comparison between the control foot force and the AMTI force plate test true value is shown in Figure 12. The tracking effect is quite consistent, which further demonstrates the feasibility of the task-oriented joint design presented in this paper. Table 4 shows that the overall performance, the load carrying capacity and energy efficiency are excellent, and the test results fulfill the design goal.

The relevant test data records are attached to the text.
(6)$$T=\frac{90\%}{60\times 8.8\%}\times 15\approx 2.56\;h$$
(7)$$COT=\frac{Energy}{mg\times Distance}=\frac{6.22\times 1000\times 3600}{510\times 9.8\times 610}\times 0.088\approx 0.646$$

## 7. Discussion

### 7.1. Simulation and Physical Prototype

We employ the Webots simulation platform, which is grounded on the open-source dynamics engine ODE, to model the interaction dynamics, drive forces, and sensor outputs of the robot with its environment. System-wide simulation is a critical step after the optimization of various subsystems’ designs, as the accuracy of its results directly influences the ultimate design outcome. In pursuit of improved realism and world consistency, we translate SolidWorks models endowed with material attributes into URDF files for simulation, where the introduced errors in mass and inertia are minimal. We also incorporate IMU errors in the simulation:(8)$$\begin{array}{cc} {\tilde{\omega }}^{b} & =\omega^{b}+b^{g}+n^{g}\\ {\tilde{a}}^{b} & =a^{w}+b^{a}+n^{a} \end{array}$$
denoted as superscripts *g* for gyroscope, *a* for accelerometer, *w* for the world coordinate system, *b* for bias random walk, and *n* for white noise. The true values of the IMU in the simulator are denoted as *w* and *a*, while the measurements are denoted as $\tilde{\omega }$ and $\tilde{a}$.

The torque commands calculated by the control algorithm are fed into the simulation engine interface through a low-pass filter, Lf(τ), of certain cut-off frequency to mimic the current closed-loop characteristics of real motor joints. A random disturbance, Rd(τ), is added to simulate the characteristics of real joint friction and fluctuations in motor torque output:(9)$$\tau_{o}=Lf\left(\tau \right)+Rd\left(\tau \right)$$

The size of this disturbance is adjusted in the simulation to be a random number between 0 and 5% ·τ. Additionally, we model a non-flat ground environment.

We employ a commercially available A1 [12] robot model for experiments, applying the same control algorithm and gait as in the simulation for comparison. By comparing the torque and speed curves in both simulation and real-world experiments, we find their trend and magnitude to be highly consistent, as illustrated in Figure 13. The error is listed in Table 5, with the root mean square (RMS) torque error at the hip joint being the largest at 8.3%. The RMS torque errors for the HAA joint and KFE joint are both below 5%. The peak torque occurs at the moment of foot-ground contact. The numerical calculation results in the simulation result in larger discrepancies leading to a larger peak torque error, but all are below 11%. The peak joint speed in the simulation is lower than that of the actual model, with the maximum error of 26.4% occurring at the hip joint, primarily due to the significant discrepancy between the inertia parameters of the hip joint in the A1 simulation model and the actual model. Given that commercial motors can significantly increase working speed by raising the input voltage, and our prototype design can provide accurate physical parameters for the simulation model, the speed error tolerance threshold in the simulation is relatively large, and the accuracy can be improved, which will not impact the subsequent design.

Based on these validations, we conclude that the system-wide simulation platform based on Webots exhibits high simulation precision, serving as a critical verification node at the concept design phase, and provides guidance for detailed design.

### 7.2. The Initial Value of Design Variables

In order to simplify the design process and quickly achieve viable design decisions, we start from a simplified 2D four-legged virtual model in Section 4 and gradually extend to a full-scale 3D simulation analysis model with detailed physical properties, where the main design decision variable is the length of the legs. The length and width of the robot’s body (measured at the intersection point of the axis lines of the hip joint and HAA joint) are set by experience. To analyze the impact of truncating the design space, we introduce seven robot models with different body lengths and widths into the full-scale simulation, keeping other physical and control parameters constant. The simulation results show that all models can walk at a speed of 2 m/s in trot gait, with particular focus on Roll and Pitch related to walking stability.

As shown in Figure 14, the simulation model with the smallest length and width (L0.6W0.2) has the largest range of Roll angle change, followed by the simulation model with the largest length and width, while the simulation model with the closest length and width has the largest range of Pitch angle change. In terms of joint torque characteristics, as shown in Figure 15, the root-mean-square joint torques of these seven simulation models are close, the peak torque of the right rear hip joint fluctuates more (the difference between the four models is within 250 N·m), and shows a characteristic that the hip-knee joint torque of the rear side is larger than that of the front side.

Considering the error of peak torque in the simulation, the instantaneousness of joint peak torque when the actual robot moves, and that the peak torque of the motor can usually reach more than three times the rated torque, we have a larger tolerance for the error of simulation peak torque in design reference. From this perspective, we can see that the change of design parameters of body length and width has less impact on the root-mean-square torque, and the initial length and width design parameters set by us will not hinder us from obtaining good joint mechanics requirements. Therefore, we further relieve the constraints on the size of other parts of the body, and the goal of design turns to integrate these subsystems into a whole with the most compact length and width after the leg-foot subsystem, drive system, and energy system are designed, and the resulting body length and width can be used as a reference for the final design.

These analysis results are based on the simulation platform and prototype model we built according to task-oriented system thinking.

## 8. Conclusions

This paper provides a detailed description of a systematic design approach for a heavy-duty and high-dynamic electrically driven quadruped robot. Following the concept of task-oriented systematic design, cascade optimization, and control-in-the-loop design, a comprehensive system integration was carried out. In the implementation process, initial designs rooted in human insights are developed, and a morphology optimization–joint-matching overall simulation analysis model is established to provide more rigorous and mathematically logical decision making for the design parameters of key components. The proposed joint drive solution, composed of fully commercialized components, was successfully applied. The experimental results demonstrate that the robot exhibits an excellent load capacity and energy efficiency, thereby verifying the effectiveness of the reported method for robot system design. Moving forward, we will focus on improving the control algorithm and conducting further tests to evaluate the robot’s performance.

## Figures and Tables

**Figure 1 sensors-23-06696-f001:**
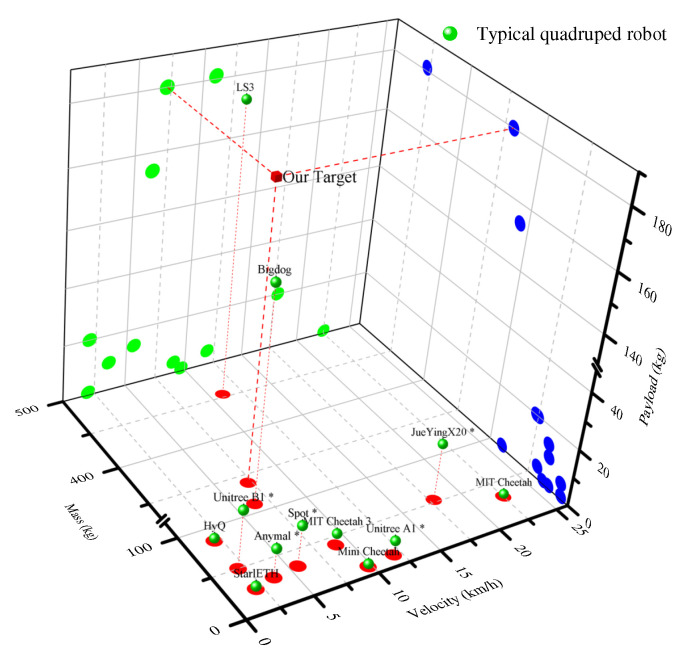
Comparative analysis of self-weight load and speed in typical quadruped robots. The robot with the star symbol in the upper right corner has been commercialized.

**Figure 2 sensors-23-06696-f002:**
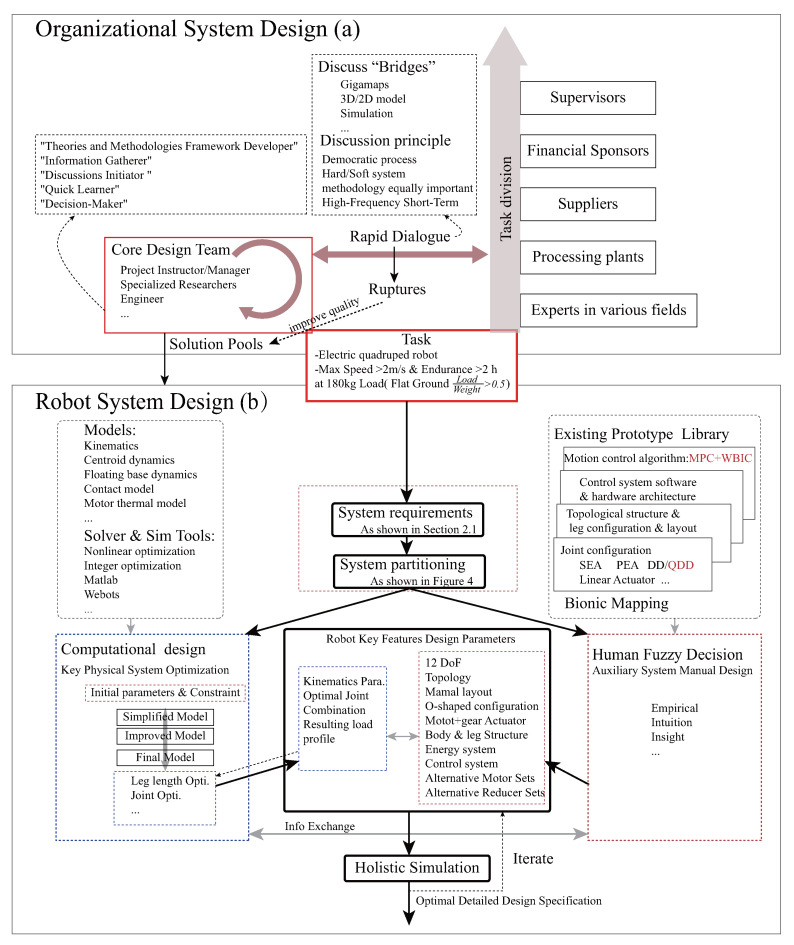
Insights into the holistic design: (**a**) Organizational system design framework. A diverse design team is assembled around the core design team. Based on task division, the team follows principles such as democracy and utilizes various tools, including GIGA-Mapping, for efficient and rapid design discussions and co-creation. As a result, an innovative and effective solution pool for the robot system design is formed. (**b**) Robot system design. Harnessing the synergy of human and machine intelligence for methodical system decomposition and design, culminating in feasible detailed design blueprints via iterative comprehensive simulations.

**Figure 3 sensors-23-06696-f003:**
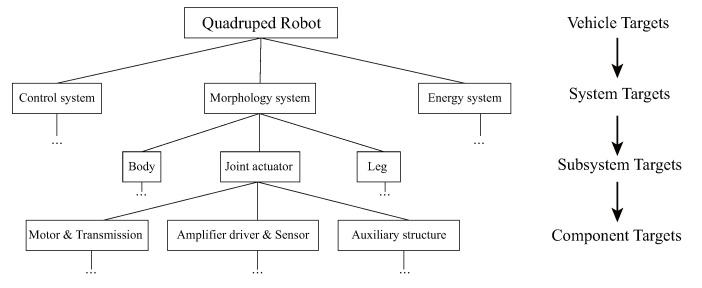
Task-based decomposition of quadruped robot system.

**Figure 4 sensors-23-06696-f004:**
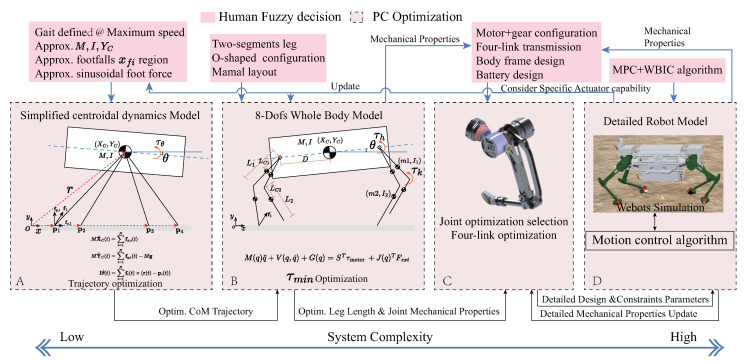
Design optimization process for key morphological parameters: (**A**) Utilizing initial estimations as inputs, the CoM and foot-end trajectories are optimized to fulfill task requirements via the centroidal dynamics approach. (**B**) Leveraging the eight-DoF float-base full-body dynamics for refining leg length to strike a balance between torque, power, speed, and inertia for an optimal outcome. (**C**) The prime component combination is selected from a set of candidates, guided by the joint torque-speed contour derived from step B. (**D**) Iterative enhancement of design parameters is conducted by evaluating the system through control system-in-the-loop simulations.

**Figure 5 sensors-23-06696-f005:**
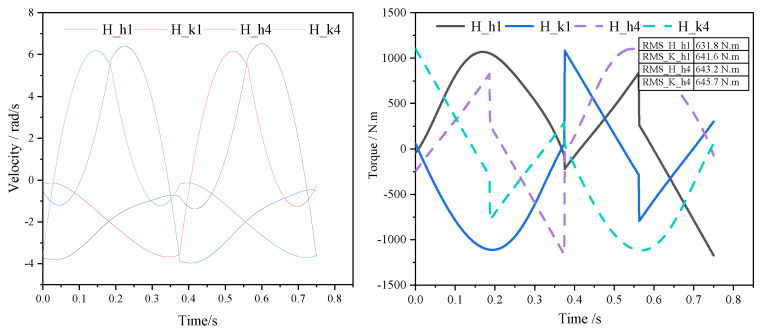
Torque−speed curve and root mean square torque for hip and knee joints derived from the optimized eight−DoF floating base dynamics model (H_h1:H(hind leg)_h(hip) or_k(knee)1(leg ID)).

**Figure 6 sensors-23-06696-f006:**
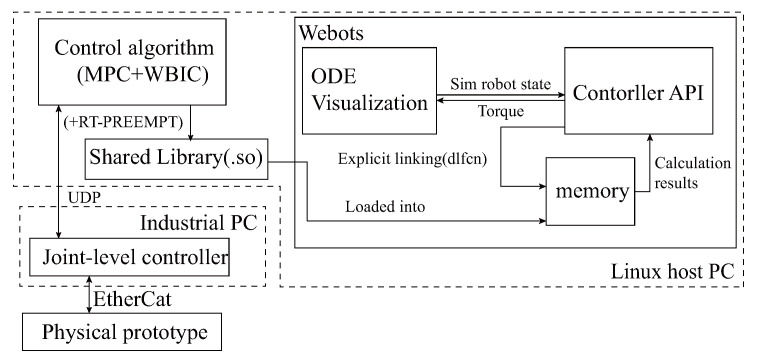
The framework for holistic simulation.

**Figure 7 sensors-23-06696-f007:**
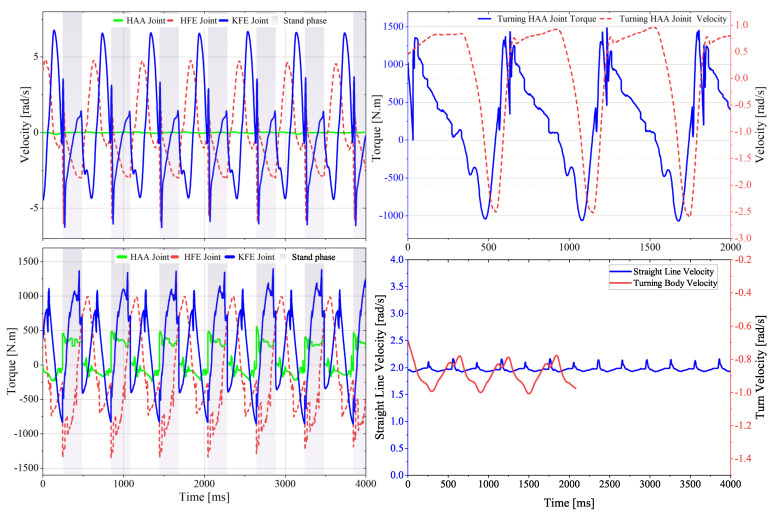
Overall simulation results: the joint torque−speed curve under extreme conditions (maximum straight−line speed load of 2 m/s) and stationary turning scenarios.

**Figure 8 sensors-23-06696-f008:**
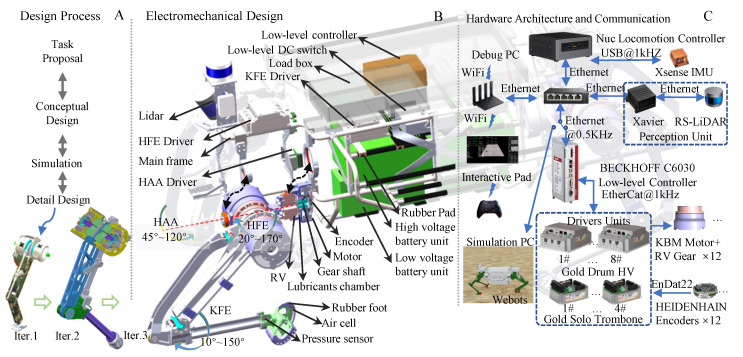
Comprehensive design brief introduction: (**A**) describes the design process and iterative version; (**B**) shows 1/4 structure because the overall layout of the robot is roughly mirror symmetrical from left to right and from front to back. It is characterized by a modular single leg and joint design, extensive range of motion, and compact structure; (**C**) sketches the hardware control architecture and simulation scheme.

**Figure 9 sensors-23-06696-f009:**
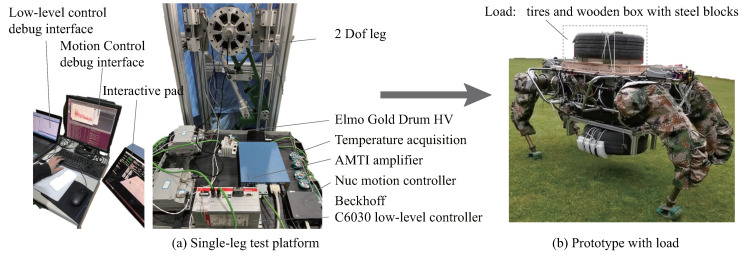
Firstly, single−leg tests (**a**) are conducted to validate joint dynamic performance and the overall electrical architecture. The validated design approaches will subsequently be incorporated into the prototype (**b**) design.

**Figure 10 sensors-23-06696-f010:**
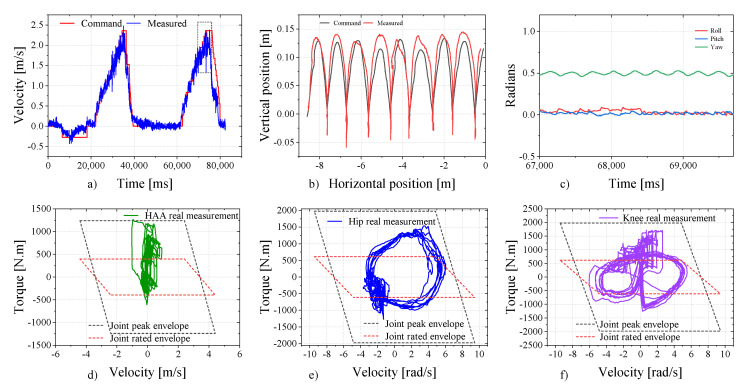
Test curve at 2 m/s speed with trot gait under 179 kg load.

**Figure 11 sensors-23-06696-f011:**
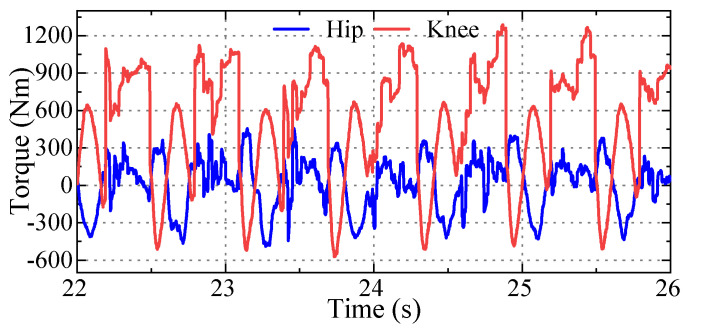
Test joint torque curve at low speed with trot gait under 210 kg load.

**Figure 12 sensors-23-06696-f012:**
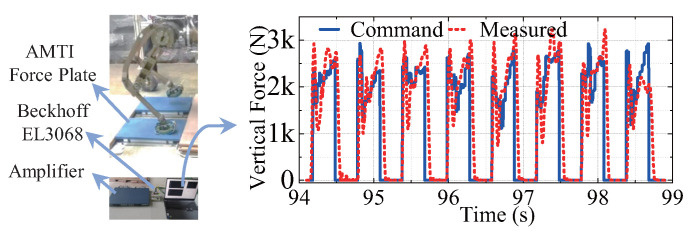
Comparison of planned command vertical force and force measured by AMTI Force Plate.

**Figure 13 sensors-23-06696-f013:**
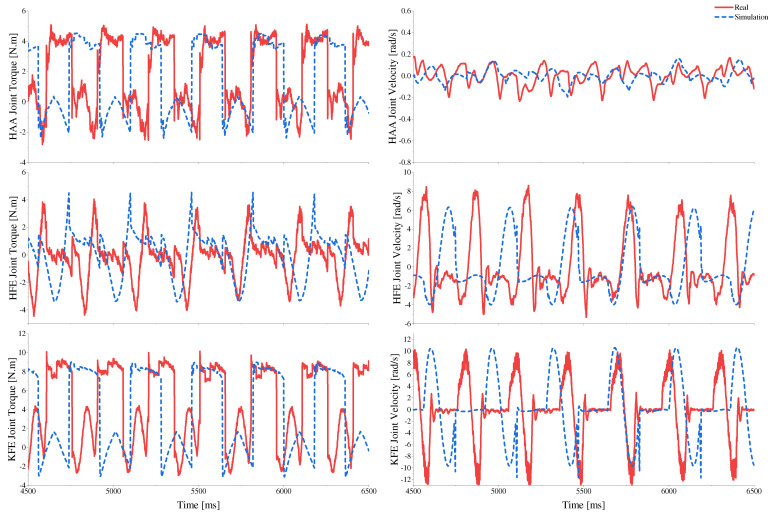
Comparison of joint torque−speed between simulated and physical prototypes under the same gait and speed conditions.

**Figure 14 sensors-23-06696-f014:**
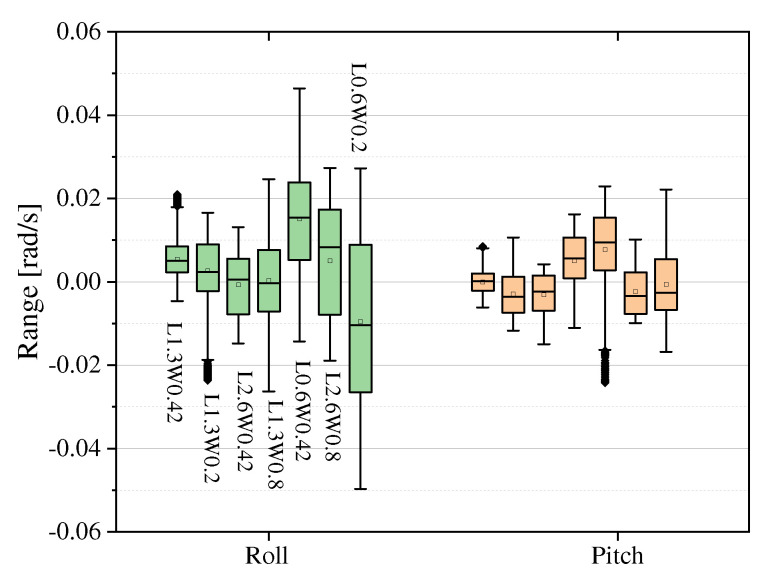
Roll and pitch variations in simulations of robots with different length−width ratios.

**Figure 15 sensors-23-06696-f015:**
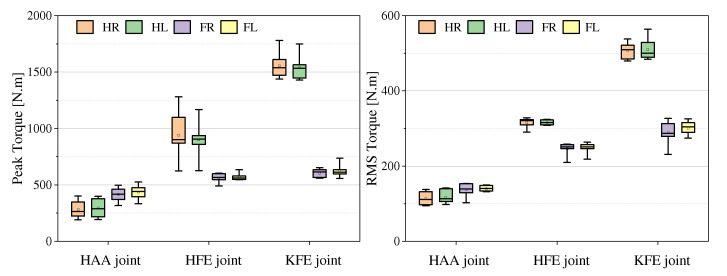
Comparative analysis of torque in simulations of robots with different length-width ratios (HR: hind right leg, HL: hind left leg, FR: front right leg, and FL: front left leg).

**Table 1 sensors-23-06696-t001:** Joint performance data derived from above condition simulation.

Joint	Rms Torque (N·m)	Peak Torque (N·m)	Peak Velocity (rad/s)	Rms Power (W)	Peak Power (W)
HAA (Max Line V/Max Turning V)	238.1/683.5	558.3/1485.0	0.2/2.6	5.8/670.0	25.8/1881.6
HFE (Max Line V)	668.9	1354.0	4.8	1145.0	3989.5
KFE (Max Line V)	655.1	1397.0	6.8	1202.9	2855.6

**Table 2 sensors-23-06696-t002:** Key off-the-shelf electromechanical parts data.

Component Name		Specifications
Actuator		
Motor+Gear+Driver	HAA	KBM43H01-A+RV25-N(81:1)+ G-SOLTROR22/400
	HFE and KFE	KBM43H03-B+RV42-N(41:1)+ G-DRUR100/400
Encoder (Heidenhain EBI 1135)		19-bit single-turn 16-bit multi-turn
Controller		
Nuc		i7@4.7 Hz, 16 G RAM; running Linux with RT patch
BECKHOFF C6930		i5@2.9 Hz; Twincat running on Win 7
Perception controller and IMU		NVIDIA Jetson AGX; MTi-G-710

**Table 3 sensors-23-06696-t003:** Key physical data of HDog.

Parameter	Value
Single leg module × 4 (weight: kg)	50 × 4
Body frame, etc. (weight: kg)	31
Controller and driver (weight: kg)	18
Battery pack (weight: kg)	90
Total (weight: kg)	339
Thigh inertia (kg·m${}^{2}$)	0.28
Shank inertia (kg·m${}^{2}$)	0.30
Dimensions (m) L×W ^a^ × H ^b^	1.55×0.42×0.95
Link lengths (m)	0.5 (Thigh and shank)
Degrees of freedom	12 (three per leg)
Leg layout	Mammal
Leg configuration	O
HAA (rotation angle)	90°
HFE (rotation angle)	160°
KFE (rotation angle)	140°

^a^ Distance between two parallel HAA axes. ^b^ Vertical distance from upper plane of body frame to sole under natural standing posture.

**Table 4 sensors-23-06696-t004:** Final test result.

Test Indicators	Result
Max velocity with load	2 m/s @ 179 kg load
Load weight ratio	0.52
Endurance	2.56 h
Other	COT = 0.646

**Table 5 sensors-23-06696-t005:** Error in joint performance between simulated and physical prototypes.

Joint	RMS Torque (N·m)			Peak Torque (N·m)			Peak Velocity (rad/s)		
	Simulation	Real	Error * (%)	Simulation	Real	Error * (%)	Simulation	Real	Error * (%)
HAA	3.066	2.959	3.5	4.601	5.117	10.1	0.470	0.494	4.9
HFE	1.588	1.466	8.3	4.538	4.227	7.4	6.35	8.63	26.4
KFE	5.808	6.088	4.6	9.029	10.129	10.9	10.82	11.16	3.1

* The error is an absolute value.

## Data Availability

Access to some complete experimental and simulation data is provided: Zenodo. https://doi.org/10.5281/zenodo.8123364, accessed on 15 June 2023.

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
