# Peer review of "Task-Oriented Systematic Design of a Heavy-Duty Electrically Actuated Quadruped Robot with High Performance"

_sensors, 2023, doi:10.3390/s23156696_

Round 1
Reviewer 1 Report
This paper proposes a task-oriented systematic design approach for a heavy-duty and high-dynamic electrically driven quadruped robot. Following the concept of task-oriented systematic design, cascade optimization, and control-in-the-loop design, a comprehensive system integration is provided in details. The experimental results demonstrate the effectiveness of the proposed method for robot system design. In general, this paper is well organized and the research work is meaningful and innovative. This paper can be accepted for publication.
Minor editing of English language required
Reviewer 2 Report
This paper presents an interesting and relevant study on the design of robotic systems. The paper reads well, is logically structured, and the authors present their work in a clear and understandable narrative.
The paper has a descriptive title, and appropriate abstract, and suitable keywords. I found the figures to be clear and of good quality. The presentation of the results is clear and I noted the availability of the data to enable the reproducibility of the research to be assessed by other related research groups. The manuscript sets out clearly the research methods, the materials and methods, the results, and conclusions.
I have comments:
1) It is recommended that the Introduction be revised in two dedicated sections as the current Introduction is a conflation of an introduction and literature review. The Introduction should be rewritten in a new Introduction (motivation and background, brief overview of the study, contribution, and paper structure) and Related Research (setting out the consideration of the related research considered along with an analysis and open research questions).
In summary, I found this to be a good paper and, while it addresses an area of research which in itself is not new, the paper provides an interesting contribution to the literature addressing robotics. There are minor revisions to the current introduction but this will be simple to implement. Upon suitable minor revision the paper will be suitable for publication in my view.
Reviewer 3 Report
In this study, the authors designed a heavy-duty quadruped robot systematically, which improved the robot in terms of velocity, load, and energy efficiency. The authors also described their design philosophy and optimization process in detail. However, several questions remain unclear, and the manuscript requires further consideration after addressing the following concerns.
1. In the abstract, from lines 19 to 23, the last two sentences are almost the same. Please revise it. An abstract is very important for a paper.
2. Spend more text explaining which one is optimized in Figure 5 and how you conclude it is optimal. No analysis regarding Figure 5; readers cannot understand it.
3. In Figure 7, what do you mean by “refine”, increased or decreased the torque peak? What is a refined velocity or torque should be?
4. Velocity = distance/time, the peak of velocity exceeding 2m/s does not mean the robot can walk at a speed of 2m/s. It is better to mention how you conclude robot can walk at a speed of 2m/s.
5. In Figure 14, specify what is HR, HL, FR, FL?
